# Nanomodification, Hybridization and Temperature Impact on Shear Strength of Basalt Fiber-Reinforced Polymer Bars

**DOI:** 10.3390/polym13162585

**Published:** 2021-08-04

**Authors:** Karolina Ogrodowska, Karolina Łuszcz, Andrzej Garbacz

**Affiliations:** Faculty of Civil Engineering, Warsaw University of Technology, UL. Armii Ludowej 16, 00-637 Warsaw, Poland; luszcz.karolina@wp.pl (K.Ł.); a.garbacz@il.pw.edu.pl (A.G.)

**Keywords:** nanomodification, hybridization, modification, fibre, nanosilica, temperature, basalt, carbon, polymer, reinforced

## Abstract

This paper presents fiber-reinforced polymer composites which were modified by fibers hybridization as well as matrix nanomodifiaction with nanosilica. The article analyzed the nanosilica matrix modification and basalt-carbon hybridization’s effect on key properties of composites use as the main reinforcement in concrete structures. The comparative analysis was based on results of bars strength parameters determined in a shear test with the ASTM standard. The tests were performed for three bar diameters at room temperature and pre-heated FRP composites at 80 °C and 200 °C for 2 h with the aim of verifying the influence of the fiber hybridization-basalt-carbon fiber-reinforced polymer (HFRP) bars and the effect of nanosilica modification of the epoxy matrix (nHFRP). The test results were also compared with results of the shear test carried out after the bars were heated to 80 °C for 30 min in order to verify and evaluate the effect of the heating time. These types of tests are relevant to the conditions that occur in FRP composites when exposed to elevated temperatures.

## 1. Introduction

In recent years, fiber-reinforced polymers (FRP) composites have been widely used in civil engineering. FRP are well-known for their high strength, stiffness, and corrosion resistance. In the composite system, strength and stiffness are mainly derived from fibers and the matrix binds fiber together [1]. Polymer matrix, together with the strengthening, must form a coherent interface, most often on the basis of an adhesive and mechanical connection, which allows the transfer of loads from the matrix to reinforcement [2]. These are the conditions for obtaining good properties of the resulting composite material. FRP’s mechanical properties are influenced by the content of the fiber fraction, the distribution of fibers in the core of bar, accuracy of the fiber coating in the resin, the type of resin, and the matrix modification [3]; therefore, a complex approach in research composite properties is extremely important.

Nowadays, new modifications of FRP composite structures are still being implemented. Modifications may concern the composite fibers, their type and arrangement, and matrix editions. Figure 1 shows that the division of FRP composites modifications depend on the type of changed component of the composite. An example of the modifications related to the composite fibers are hybrid-fiber polymers composites (HFRP). Polymer-hybrid composites were created as a result of striving to obtain higher and higher properties of materials. Combining fibers with different properties allows us to use the advantages of individual fibers, their mechanical and physical properties, and allows for a more rational use of the material in terms of cost. In hybrid composites, different fibers in the cross-section of the composite are used. For example, in basalt fiber bars (BFRP), we can use carbon fibers in the core to increase the stiffness of BFRP bars by replacing a proportion of basalt fibers with carbon fibers and also the modulus and strength [4]. The fatigue performance of glass-reinforced polymers (GFRP) can also be improved by adding carbon fibers. The high performance that hybrid composite material offers led to their continued development [1,3].

Fiber-reinforced polymer composite matrices can also be modified by the addition of different kinds of fillers, for example, silica, rubber, zirconium, zinc oxide, glass, ceramic, metal, and other particles. The type of selected fillers increases some expected properties but overly large particles and the non-suitability of content in the resin matrix can significantly increase the viscosity of resin. High viscosity reduces processing of the mixture and can lower the basic mechanical properties of the cured epoxy binder. Nano-sized particles are preferred [5,6]. Nano-fillers have become an issue of research and development for polymeric materials. Their large, specific surface is significant for the interaction effect with the matrix particles and has a positive influence on processing and the final properties of a composite. The most widely investigated additive is nanosilica. This type of particle is regarded as a reference filler material. Its compact spherical structure is unchanging under the stress exerted on it in melt during flow. It is used as a filler in polymeric materials to reduce the permeability to gases or to improve flame resistance by increasing the glass transition temperature (Tg) [5]. It is extremely important in FRP usage, like main reinforcement in construction because one of the main problems is the sensitivity of the resin to high temperatures. Temperature changes not exceeding the Tg value do not cause adverse effect on the epoxy matrix. However, if the value of the glass transition temperature is exceeded, the strength parameters are significantly reduced, which is related to the transition of the resin from the glassy to the rubbery state [3,7].

The FRP bars applicable for the reinforcing of concrete structures are mainly ex-posed on the temperature of seasonal and daily temperature variations which impact construction. Usually, is it the normal range below glass transition temperatures for composites in which they retain their original properties. However, it is expected that the FRP bars in the structure will have a certain load capacity under fire conditions. H. Hajiloo et al. [8] investigated the behavior of bending members, especially simple concrete slabs-reinforced FRP bars. During the fire tests, one of the key elements was to ensure adequate bar-concrete adhesion in the middle of the plates. Insufficient bar-concrete adhesion in the middle of the slab was the reason for the failure of ones of the first full-scale tests [9,10]. A study by E. Nigro et al. [10] showed the effect of cold-anchoring zones on the performance of GFRP-reinforced slabs. Experience has shown that failure of FRP concrete slabs will occur in the center of the element in tension if there is a sufficiently long area that is not exposed to fire on the supports. The failure of the slab was caused by insufficient adhesion in the area above the supports when the temperature in the unexposed area exceeded the glass transition temperature. Appropriate design of supports and the use of high Tg resin can ensure better safety of the structure when exposed to high temperatures.

The aim of this paper was investigation of temperature and heating-time effects on the mechanical properties determined in a shear test method in accordance with ASTM [11]. We investigated the three diameters of three types of bars developed at the Warsaw University of Technology: BFRP (basalt fiber-reinforced polymer) and HFRP (hybrid fiber-reinforced polymer) to increase the stiffness of the BFRP bars by replacing 25% of the basalt fibers with carbon fibers and nHFRP-hybrid FRP containing an epoxy matrix modified with 3% SiO_2_ nanosilica to research the impact this modification has on nHFRP’s high temperature resistance. Each type of bar (BFRP, HFRP, nHFRP) were tested pre-heated to 80 °C and 200 °C for 2 h and at room temperature. The temperature of 80°C is the most universal in the operational state and oscillation around the glass transition temperature of the investigated composite. The temperature of 200 °C was selected as the probable temperature obtained inside the concrete element, and in the anchorage zone in the FRP bars were covered with concrete during the exposure over the source of fire. After heating, the bars were stabilized at room temperature, reweighed, the surface condition was analyzed, and then the tests were started. As a control test of the influence of elevated temperature, the transverse shear tests were realized in accordance with the ACI 440.3R-04 guidelines [11], which is a quicker test than the most often-used FRP research-tensile test. It does not require any special preparation of samples-special anchorage, while on its basis it can be concluded that some factors have a negative effect on the structure of the bar and its mechanical properties.

## 2. Literature Review of Nano Silica Modification and Hybridization Impact on FRP Composites

The significant factor for operating and fire-resistant characteristics of FRP bars used is Tg (transition glass temperature). Obtaining the highest possible glass transition temperature is directly related with maintaining their properties for as long as possible, regardless of changes in ambient temperature. Thermal properties and the increased Tg of epoxy polymers can be improved by adding SiO_2_ nanoparticles to the polymer matrix [12]. The inorganic nanoparticles added to the resin reduce the segmental mobility of the polymer significantly, affecting the glass transition temperature. Several different thermal analysis methods can be used to determine the glass transition temperature. With the fact that the indicated methods use measurements of various physical values, the determined glass transition temperature, depending on the method, may be different [13]. Based on the data [12,14,15,16,17,18,19,20], a graph was developed which defines the influence of the amount of added nanosilica on the obtained glass transition temperature. Some of the tested epoxy resins showed a significant increase in Tg and in some cases there was a decrease (Figure 2).

The literature analyses have shown that there are no generally accepted methods for modifying epoxy resin with nanosilica. The resins and silica of various origins were used in the research [12,14,15,16,17,18,19,20]. The curing system, the manner of adding nanofiller to the epoxy resin, as well as the mixing and hardening times were also different. All these factors influence the shaping of the individual properties of both the composite mixture and the hardened composite. The authors of articles have been looking at the individual ways to improve specific properties, which are not always the same, so the selected methods and means used in the research have been different. These methods included, among others applications, nanofiller surface modification.

The surface modification of nanosilica is to help the dispersion of silica nanoparticles in the epoxy resin. Nanoparticles can be added by physical or chemical modification. Particles prevent agglomerates and are homogenously arranged in the epoxy matrix. Chemical methods consist of modifying silica with low molecular weight compounds or by grafting polymers. The most popular chemical modification agents are silanes. Silanes make nanosilica particles hydrophobic, and therefore more favorably interact with polymers [21]. Due to the fact that epoxy resins are characterized by relatively high viscosity, when adding nanosilica to the binder, in a fairly large number of the works various diluents were used when mixing the ingredients. The solvent is necessary when the addition of nanosilica causes a large increase in the viscosity of the epoxy resin, making freely mixing impossible. Thanks to its addition, it is possible to more evenly distribute the nanofiller in the epoxy matrix. In literature analyses [12,14,15,16,17,18,19,20], the most-used solvent was acetone. Another factor to obtain the dispersion of particles immediately in mechanic or magnetic mixing is ultrasonication. The ultrasounds improve the homogeneity of the composite mixture. In the mixture of resin and nanosilica exposed to ultrasound, the cycles of high pressure (compression) and low pressure (dilution) processes in the liquid are important. This leads to the production of a liquid stream under high pressure that breaks up the agglomerated particles [22]. The epoxy resin with nanosilica mixing using the ultrasound is a good alternative to classic mechanical mixing, providing better particle dispersion, especially with a higher percentage of nanosilica compared with epoxy resin. The process of nanomodification with silica is quite complicated, as it requires the establishment and constant control of many initial conditions in an appropriate way to obtain the expected properties.

Another type of modification of FRP composites is fiber hybridization. The obtained properties of the hybrid composite are influenced by many factors, e.g., the order in which the fibers are arranged in the composite. A. Subagia et al. [23] investigated the influence of the sequence of laying the layers of carbon and basalt fabrics on the properties of the resulting laminates. The results showed that the flexural strength and modulus of elasticity of the hybrid composite laminates significantly depended on the order in which the fibers were laid. However, all stacking configurations showed a positive hybridization effect. The arrangement of the carbon fibers on the compression side resulted in a greater strength and modulus of elasticity than when the basalt fabric was placed on the compression side [23]. This also affects the results obtained for bar-shaped composites and the effect of fiber distribution in the bar core.

Hybridization may concern mixing basic fibers glass, basalt, carbon, and aramid in the cross-section of the bar, but is also reflected in the innovation solutions. For example, in types of composites with a large rupture strain (FRP LRS), in contrast with traditional FRP composites, it is expected that an increase rupture strain leads to a better performance of structures [24,25]. Another new type of composite bar is the steel-fiber-reinforced polymer (SFCB) made of an inner steel bar and outer FRP, integrated in a pultrusion process. The properties of this type of hybrid composite are determined by many factors, for example, the steel/fiber ratio, type of used fibers (carbon, basalt), and the bond strength between steel and fibers [26]. Summarizing, hybridization is a predictable modification, directly affects the mechanical properties of the resulting composite, and new solutions are still being implemented in order to rationalize costs and material consumption.

## 3. Materials and Methods

Shear tests of FRP were carried out on three types of bars, BFRP, HFRP, and nHFRP, with diameters of 4, 8, 10 mm, in which the volume fraction of resin was 20% and the fiber fraction 80%. In order to increase the elasticity modulus, and thus increase the stiffness of the bars, BFRP bars were hybridized with carbon fibers in the core (Table 1).

The FRPs with the epoxy binder modified with nanosilica were analyzed. An epoxy binder system consists of the following components:−component A: low molecular weight epoxy resin obtained from bisphenol A and epichlorohydrin;−component B: anhydride-type hardener;−component C: accelerator in the form of a tertiary aliphatic amine, used as an agent accelerating the epoxy resin crosslinking process;−component D: modifier, polypropylene glycol diglycide ether, used as an active diluent to make the resin elastic.

The system was developed to obtain structural composites using the pultrusion method. According to the manufacturer’s recommendation, the individual components were mixed with a mechanical mixing in the proportion A:B:C:D = 100:70:5:7 (weight ratio). Additionally, a second type of epoxy resin was prepared with the addition of nanosilica. The size of the nanosilica used as a modifier was determined using a Mastersizer device (Malvern Instruments Ltd., Malvern, UK). The average size of the nanosilica used was 24.37 nm, in which two fractions were distinguished: a finer one with a peak at 30 nm in diameter (approx. 80%) and a coarse-grained one with a peak at a diameter of 1270 nm (approx. 20%). The bars with unmodified resin were marked as HFRP in the tests, while the bars with resin binder with the addition of nanosilica were marked as nHFRP.

Before proceeding to the shear test, all samples were cut into 300 mm sections: five samples for each type of bar, diameter, and each temperature, totaling 135 samples. Next, they were measured, weighed, and the equivalent diameter of the bar was determined. The surface assessment of the bars showed some imperfections (Figure 3), including uneven resin coverage as well as the cross-section of the bar having an asymmetric offset of the carbon fibers from the core.

The first part of the samples, 45 specimens, were pre-heated to 80 °C for 2 h, and the second part, 45 samples, were pre-heated to 200 °C for 2 h. The specimens were next stabilized at room temperature, reweighed, the surface condition was analyzed, and then the tests were started. A third part of the tested samples, 45 specimens, remained at room temperature all the time.

The transverse shear tests were realized in accordance with the ACI 440.3R-04 guidelines [11]. The shear test used a testing machine Instron 3382 Floor Model Universal Testing System with a loading capacity in excess of the shear capacity of the test specimen and was calibrated according to the ASTM standard (Figure 4). The machine was in the typical test setup. It was consisted of a sample holder, one upper blade, and two lower blades. The shear testing device was constructed of steel. During the test, the bar-shaped specimen was sheared on two planes simultaneously by the blades which were converging along faces perpendicular to the axis of the test specimen. The sample holder had adequate dimensions and a characteristic longitudinal V-shape cut for placed FRP samples and a rectangle cut for hold upper and lower blades in the center of its top part (Figure 4).

During the test, bars was mounted in the center of the shear apparatus, touching the upper loading device (Figure 4). The load rise rate applied to each sample was 50 MPa/min. Load was applied uniformly without shock. A temporary decrease occurred in some samples was occurred due to the presence of two rupture faces. The FRP bars were tested to obtain the maximum shear force. The shear failure was determined by visual inspection. The shear strength was calculated according to Equation (1).
τ_u_ = P_s_/2A(1)
where: τ_u_—shear strength, MPa; Ps—maximum failure load, N; A—cross-sectional area of the specimen, mm^2^.

## 4. Results and Discussion

### 4.1. Effect of Nano Silica Modification, Basalt-Carbon Hybridization and Elevated Temperatures

The results of the shear test are presented in the summary table for all tested diameters and types of FRP composites (Table 2). The average shear strengths for monofilament bars, BFRP, with carbon-basalt bars, HFRP, were compared to evaluate effect of the hybridization (Figure 5). In the same way, we compared unmodified bars, HFRP, with nHFRP bars with nanosilica to show the influence of nanomodification on the strength parameters of composite, as well as the reaction to pre-heating (Figure 6).

Due to the influence of temperature, there was a decrease in the average shear strength for BFRP bars, regardless of the diameter. A decrease in the range of 0–5% for bars which were heated to 80 °C was observed, while a decrease above 5% for composites heated to 200 °C, which can be considered as visible temperature sensitivity but on a small level, was also observed. For HFRP Ø4 bars, both at temperatures of 80 °C and 200 °C, values fell to 5%, so the temperature effect can be considered insignificant. For the HFRP bar Ø8, the decrease was around 10% at 80 °C and 200 °C. The HFP Ø10 bar proved to be highly sensitive to elevated temperature, and tests on heated bars showed a decrease of 29.70% at 80 °C and 36.70% at 200 °C. The FRP composites-modified nanosilica-nHFRP Ø4 showed a decrease and growth of about 5%, which can be considered as a neutral effect of temperature, taking into account the obtained results of the standard deviation for the value of the mean shear strength. For bars nHFRP Ø8, an increase in the average strength value was obtained, about 5% for 80 °C and 7% at a temperature of 200 °C. The strength increase, especially at 200 °C, may be caused by post-hardening of the resin after it stabilizes. For the heated nHFRP Ø10 bars, there was a decrease in the average shear strength in the range of 17–21%.

The hybridization of bars tested at room temperature had a positive effect for each researched diameter (Figure 5). An increase the average shear strength in relation to unmodified bars: Ø 4—about 19%, Ø 8—about 12%, and Ø10—about 13%. However, the imperfections of this modification were revealed by heating the bars Ø10 to the temperatures of 80 °C and 200 °C. The obtained results were about 17% heated to 80 °C and 23% heated to 200 °C, more reduced than the unmodified bars (BFRP). Nevertheless, the heating did not have an overall negative effect on the hybridized bars, because for diameters Ø4 and Ø8, the increase in strength parameters of the HFRP bars compared to the unmodified ones was maintained at the level of about 15%. Hybridization has been shown together with the effect of temperature due to the possibility of revealing some imperfections of the bars after heating.

The FRP bars with the nanomodified matrix had, at room temperature, a negligible impact on the strength parameters of the bars in relation to the unmodified bars, in a range less than 5% (Figure 6). On the other hand, for Ø4 nHFRP bars, the decrease of average shear strength at room temperature in relation to HFRP was 24%, and about 10% after heating. A different phenomenon occurred in the nHFRP bars Ø10, which showed an increase of shear strength of 9% after heated to 80 °C for 2 h and 25% after heated to 200 °C. The influence of the modification performed was analyzed with influence of temperature due to the monitoring of the behavior of nanosilica during the preheating process.

The FRP bars’ damaged surface (Figure 7) showed many differences in the behavior depending on types and bar diameter. Some of the bars, after the destruction, had a characteristic smell of burning, and the surface of the fracture showed an orange/yellow discoloration, indicating that the resin was partially burned. A poorly bonded Ø10 bars structure (matrix-fibers) could have been caused a visible delamination of their microstructure after bar damage. The delamination of the composite increased with the taking of loads from the testing machine, which led to a faster development of failure. Two types of cracks were observed: (1) violent and loud brittle fracture; (2) less-rapid destruction: the fibers did not brittle, but only bent towards the applied force.

The delamination resistance depends primarily on the properties of the resin and the adhesion between the fiber and the resin. Prevention of delamination is achieved by increasing the resin volume in the FRP bar and uniform matrix coverage of the fibers. In the future, it is very important to systematize the properties of composites and microstructural approach in the analysis of the results of FRP composite researches.

### 4.2. Influence of the Heating Time

The results obtained in the shear test for bars heated for 2 h at the temperature of 80 °C were compared with the previous results of the shear test performed after heating the bars at the temperature of 80 °C for 30 min [3]. The differences in the transferred shear force and failure displacement were insignificant, while the heating time significantly reduced the failure time of the bar (Table 3), with time reduction ranging up to 42–52%. The test results showed reduction failure times for each type of bar: BFRP, HFRP, and nHFRP. This means that the composite destruction occurs much more rapidly when the bar is pre-heated. This reduce-time phenomenon should be included during the design of concrete elements reinforced with FRP bars. In the course of research, an exclusive focus on force reduction is not a desirable approach.

With the extension of the bar heating time, the nature of the load-displacement curves changed significantly (Figure 8). When the bars were heated to 80 °C for 30 min, the character of the curves was smooth, while after heating for 2 h at the same temperature, the load-displacement curve became less smooth, and jumps were visible.

## 5. Conclusions

The main conclusions:The hybridization had a positive effect on the shear strength parameters of the tested composites. At room temperature, the increase of the HFRP bars shear strength were higher, in the range of ~11–19% compared with the BFRP bars.The nanomodification of epoxy resin with silica is a promising way for matrix modifications to improve resistance in elevated temperatures. For the Ø10 nHFRP bars, an increase in shear strength was observed in relation to the unmodified bars, HFRP, for bars pre-heated to 80 °C, 9% and 25% pre-heated to 200 °C.The effect of temperature variation is the most visible for the biggest diameters of hybrid FRP. The effect was less noticeable for single-type fiber bars, BFRP.The heating time had a significant impact on the shape of the load-displacement diagram-change from a smooth line to a line with jumps, and also significantly shortened the failure time. In the tests, the failure time of bars heated to 80 °C for 2 h compared with bars heated for 30 min was reduced by the range of 42–52%.The jumps on load-displacement curves were more visible for the BFRP bar type and less for hybrid FRP bars. No significant effect of nanosilica was observed.

## Figures and Tables

**Figure 1 polymers-13-02585-f001:**
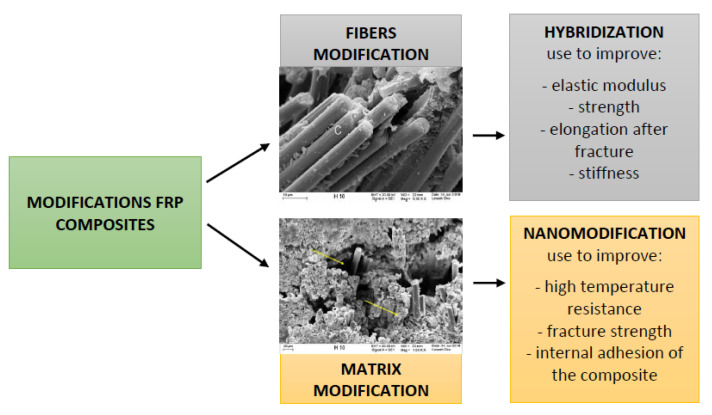
Modification scheme of individual components of the composite-fibers and matrix, and its potential impact on the main final properties of the resultant material.

**Figure 2 polymers-13-02585-f002:**
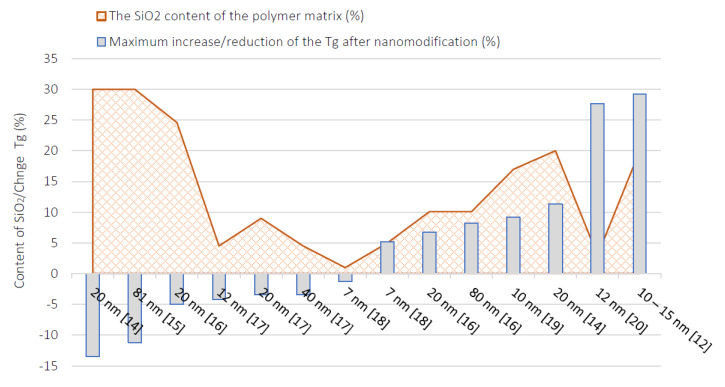
The glass transition temperature (Tg) of obtained FRP composites for different contents of SiO_2_. Literature review [12,14,15,16,17,18,19,20] for different diameters of nanosilica: 7–81 nm.

**Figure 3 polymers-13-02585-f003:**
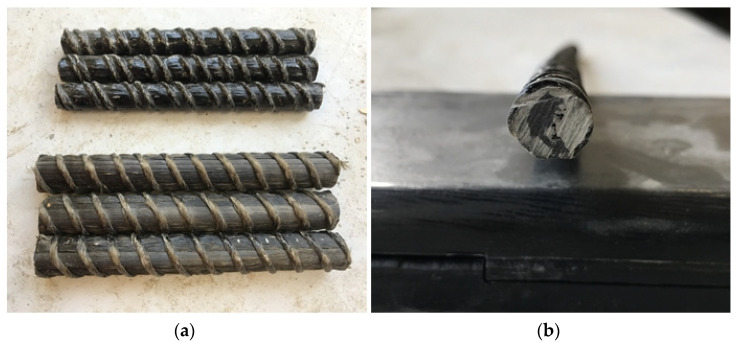
FRP bar imperfections: uneven coverage of epoxy resin on surface of bars (**a**) variable distance of the carbon core from the bar edges (**b**).

**Figure 4 polymers-13-02585-f004:**
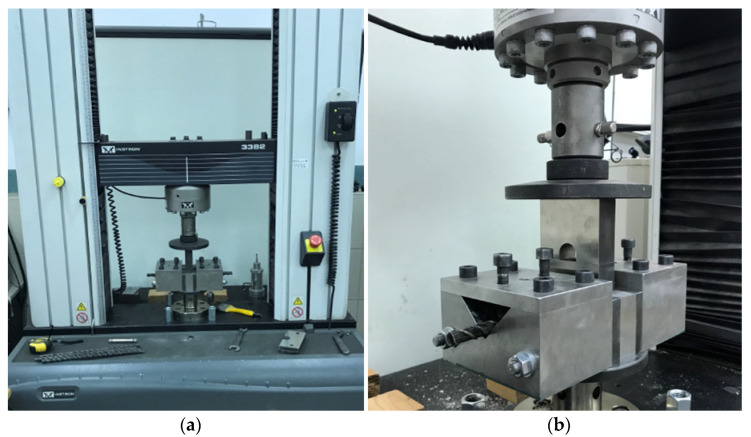
The shear test: (**a**) shear machine mounted in the testing machine; (**b**) the sample holder with a longitudinal V-shape cut for placed FRP.

**Figure 5 polymers-13-02585-f005:**
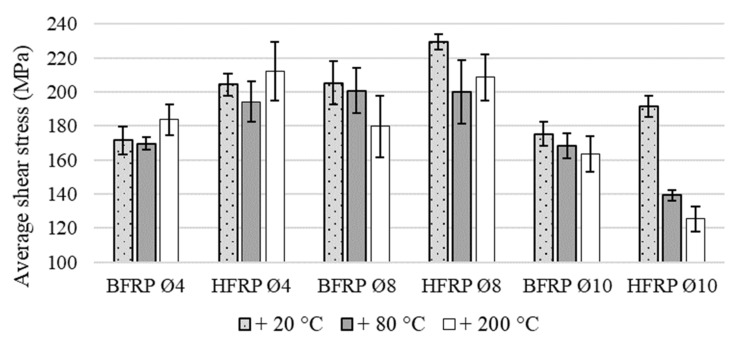
Comparison of the average shear strength for BFRP (basalt fiber-reinforced polymer) with HFRP (hybrid fiber rein-forced polymer) with average shear strength standard deviation (SD).

**Figure 6 polymers-13-02585-f006:**
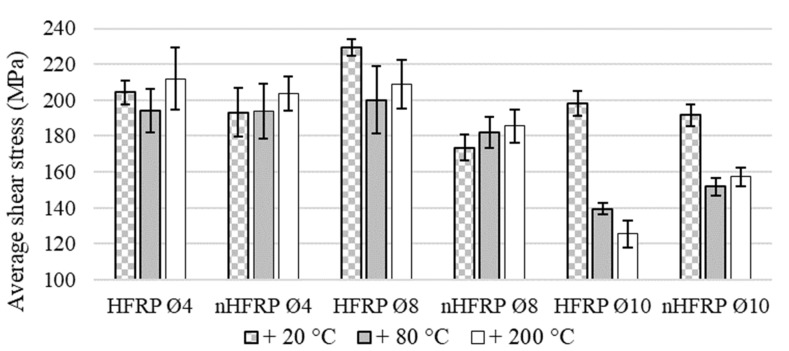
Comparison of the average shear strength for nanosilica modified bars, nHFRP, and unmodified HFRP with average shear strengths, standard deviation (SD).

**Figure 7 polymers-13-02585-f007:**
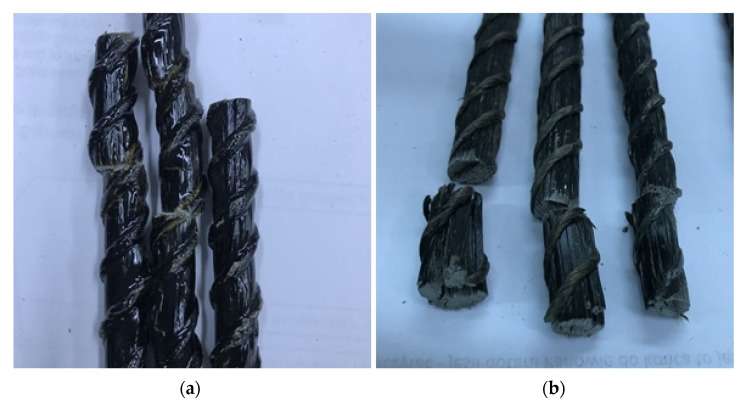
Surface changes observed after the shear test. Sample Ø8 previously heated to 200 °C for 2 h: yellow/orange color visible at the shear point (**a**), delamination of the Ø10 bar structure (**b**).

**Figure 8 polymers-13-02585-f008:**
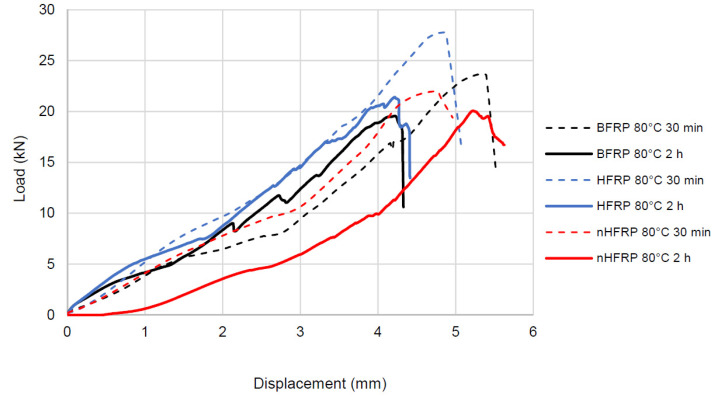
Scheme of the dependence of the shear force on the shear displacement of the press actuator for sample no. 1, diameter 8 from each type of bar: BFRP, HFRP, and nHFRP.

**Table 1 polymers-13-02585-t001:** Types of tested bars BFRP (basalt fiber-reinforced polymer), HFRP (hybrid basalt-carbon fiber-reinforced polymer), and nHFRP (hybrid bars modified nano silica).

Type of Bar	Basalt Fiber Content in the Fiber Fraction	Carbon Fiber Content in the Fiber Fraction	Content SiO_2_ in the Matrix
BFRP	100%	-	-
HFRP	75%	25%	-
nHFRP	75%	25%	3%

**Table 2 polymers-13-02585-t002:** Shear test results for Ø4, Ø8, and Ø10. Tested bars: BFRP (basalt fiber-reinforced polymer), HFRP (hybrid fiber-reinforced polymer), and nHFRP (hybrid bars modified nano silica).

Type of Bar	Pre-Heating Temperature (°C)	Equivalent Diameter (mm)	Average Shear Force (kN)	Average Shear Stress (MPa)	CV Average Shear Stress (%)
BFRP Ø4	20	4.64	5.798	171.52	4.66
	80		5.733	169.61	2.21
	200		6.208	183.65	4.93
HFRP Ø4	20		6.901	204.41	3.21
	80	4.64	6.555	194.18	6.13
	200		7.157	211.99	8.15
nHFRP Ø4	20		7.340	193.25	7.14
	80	4.92	7.360	193.79	7.79
	200		7.740	203.80	4.65
BFRP Ø8	20	8.32	22.348	205.37	6.30
	80		21.856	200.84	6.56
	200		19.574	179.87	10.08
HFRP Ø8	20		27.726	229.44	1.98
	80	8.77	24.183	200.12	9.32
	200		25.208	208.60	6.50
nHFRP Ø8	20		20.214	173.48	4.21
	80	8.61	21.192	181.88	4.71
	200		21.612	185.48	4.88
BFRP Ø10	20	9.97	27.398	175.33	4.12
	80		26.291	168.25	4.27
	200		25.557	163.55	6.50
HFRP Ø10	20		38.245	198.31	3.48
	80	11.08	26.887	139.41	2.25
	200		24.209	125.53	5.91
nHFRP Ø10	20		35.768	191.64	3.21
	80	10.89	28.334	151.81	3.14
	200		29.369	157.35	3.29

**Table 3 polymers-13-02585-t003:** The aaverage failure time and average shear stress in relation with bar heated time; (1)—bars heated at 80 °C for 30 min, (2)—bars heated at 80 °C for 2 h.

Type of Bar	Average Failure Time (s)	Average Shear Stress (MPa)	Average Shear Displacement (mm)
BFRP(1)	494.16	208.22	5.20
BFRP(2)	233.60	200.84	5.06
HFRP(1)	540.44	226.98	4.96
HFRP(2)	229.40	200.12	4.97
nHFRP(1)	462.68	194.84	4.82
nHFRP(2)	242.60	181.88	5.52

## Data Availability

The data presented in this study are available on request from the corresponding author.

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
