# Peer review of "Nanomodification, Hybridization and Temperature Impact on Shear Strength of Basalt Fiber-Reinforced Polymer Bars"

_polymers, 2021, doi:10.3390/polym13162585_

Round 1
Reviewer 1 Report
the paper describes the preparation of fiber reinforced polymer composites and compares some mechanical properties between composites containing different kinds of fibers.
in my opinion, to explain the different behaviour observed, some considerations on the nature of the interactions originating between the chemically different fibers and the matrix could be useful. A study carried out with scanning electron microscopy on the original materials, after heating and after deformation could support and explain the results. The use of silica fibers functionalyzed with aminopropyltrimethoxysilane could significanly improve the strength of the interaction between the fibers and the matrix acting the aminogroup of the silane derivative as hardener for the epoxy precursor.
Author Response
Thank you for your answer very much.
Thank you for the very important considerations regarding interactions originating between the chemically different fibers and the matrix. At this stage, I did not perform scanning microscopy after heating and after deformation. I am sure that your comments are very important, but I do not have the possibility to do a SEM analysis now.
Best regards,
Karolina Ogrodowska
Reviewer 2 Report
I can recommend the paper "Nanomodification, hybridization and temperature impact on shear strength of basalt fiber reinforced polymer bars” for the publication in the journal “ Polymers”.
In this paper was experimental study on the fiber-reiforced polymer composites.
Polymer hybrid composites were created as a result of striving to obtain higher and higher properties of this materials.
For studies of this systems used the mechanical and thermal of these composites.
In the paper shows the improvement of the temperature and mechanical properties of these composite materials.
I think this paper will be interesting for readers of this journal .
I am recommending to include in the references the next publications:
1.S.P. Repetsky, I. G. Vyshyvana,Y.Nakazawa,S. P. Kruchinin, S.Bellucci., Electron transport in carbon nanotubes with adsorbed chromium impurities.Materials (2019), 12, 524.
2.Vlaskina S., Kruchinin S.. Rodionov V., Nanostructures in silicon carbide crystals and films Inter.J.Mod.Phys. B. vol.30, N13 ,p.1042015 (2016).
Author Response
Thank you for your accourate comments and suggestions.Thank you also for recommending me interesting publications.
I will keep them in my mind during a literature review for my future paper.